# Systematic Studies of the Circadian Clock Genes Impact on Temperature Compensation and Cell Proliferation Using CRISPR Tools

**DOI:** 10.3390/biology10111204

**Published:** 2021-11-18

**Authors:** Yue Wu, Tian Tian, Yin Wu, Yu Yang, Yunfei Zhang, Ximing Qin

**Affiliations:** 1Department of Health Sciences, Institutes of Physical Science and Information Technology, Anhui University, Hefei 230601, China; wuyue2018gd@163.com (Y.W.); tian_tian0386@sina.com (T.T.); wuyinst@163.com (Y.W.); yangyu19910725@163.com (Y.Y.); 2Moeden Experiment Technology Center, Institutes of Physical Science and Information Technology, Anhui University, Hefei 230601, China

**Keywords:** circadian rhythm, clock genes, temperature compensation, genetic editing, CRISPR

## Abstract

**Simple Summary:**

One of the major characteristics of the circadian clock is temperature compensation, and previous studies suggested a single clock gene may determine the temperature compensation. In this study, we report the first full collection of clock gene knockout cell lines using CRISPR/Cas9 tools. Our full collections indicate that the temperature compensation is a complex gene regulation system instead of being regulated by any single gene. Besides, we systematically compared the proliferation rates and circadian periods using our full collections, and we found that the cell growth rate is not dependent on the circadian period. Therefore, complex interaction between clock genes and their protein products may underlie the mechanism of temperature compensation, which needs further investigations.

**Abstract:**

Mammalian circadian genes are capable of producing a self-sustained, autonomous oscillation whose period is around 24 h. One of the major characteristics of the circadian clock is temperature compensation. However, the mechanism underlying temperature compensation remains elusive. Previous studies indicate that a single clock gene may determine the temperature compensation in several model organisms. In order to understand the influence of each individual clock gene on the temperature compensation, twenty-three well-known mammalian clock genes plus *Timeless* and *Myc* genes were knocked out individually, using a powerful gene-editing tool, CRISPR/Cas9. First, *Bmal1*, *Cry1*, and *Cry2* were knocked out as examples to verify that deleting genes by CRISPR is effective and precise. Cell lines targeting twenty-two genes were successfully edited in mouse fibroblast NIH3T3 cells, and off-target analysis indicated these genes were correctly knocked out. Through measuring the luciferase reporters, the circadian periods of each cell line were recorded under two different temperatures, 32.5 °C and 37 °C. The temperature compensation coefficient Q_10_ was subsequently calculated for each cell line. Estimations of the Q_10_ of these cell lines showed that none of the individual cell lines can adversely affect the temperature compensation. Cells with a longer period at lower temperature tend to have a shorter period at higher temperature, while cells with a shorter period at lower temperature tend to be longer at higher temperature. Thus, the temperature compensation is a fundamental property to keep cellular homeostasis. We further conclude that the temperature compensation is a complex gene regulation system instead of being regulated by any single gene. We also estimated the proliferation rates of these cell lines. After systematically comparing the proliferation rates and circadian periods, we found that the cell growth rate is not dependent on the circadian period.

## 1. Introduction

Circadian rhythms play important roles in maintaining homeostasis in mammals. The circadian clock system is a hierarchical system that incorporates external environmental cues and internal autonomous rhythms [1,2]. The circadian rhythms widely exist in almost every organ and cell in mammals. The suprachiasmatic nucleus (SCN) of hypothalamus, as the central pacemaker, regulates the biological rhythm of peripheral organs and synchronizes the peripheral clocks through complex signal cascade responses [3,4]. This timing system is very important for normal physiology and behavior. One of the major characteristics of the circadian clock is that it is temperature compensated [5]. In order for organisms to adapt to changes in the external environment to maintain normal physiological functions, changes in temperature within a certain range will not produce much change in the circadian rhythm of the organism, which is called temperature compensation [6]. Having a temperature-compensated circadian clock brings many benefits to animals, even to endothermic animals. However, the mechanism underlying temperature compensation remains elusive. 

Q_10_ is used to represent the temperature dependence of a biochemical reaction process. The range of Q_10_ in circadian systems is around 0.8–1.2 [7], which indicates the system is temperature compensated. A number of mutation studies have investigated the molecular basis of temperature compensation in model organisms. Both *F**rq* mutants in Neurospora and *Per* mutants in Drosophila have been shown to largely compromise temperature compensation [8,9], indicating that a single clock gene may directly determine the properties of temperature compensation. While in mammals, the *Tau* mutation in hamster (mutations in *Csnk1e* gene that encodes CK1ε) adversely affects the temperature compensation with Q_10_ value 1.487, compared with Q_10_ 1.096 in wild types [10]. Furthermore, in vitro biochemical phosphorylation activity of CK1δ and CK1ε are temperature-insensitive [5,11,12], indicating temperature compensation may be a property derived from biochemical reactions by clock kinases CK1δ and CK1ε. For the simplest model organism cyanobacteria, the temperature compensation is determined by the ATPase activity of its key clock protein KaiC [13,14]. These studies indicate that the temperature compensation may be affected by a single clock gene and/or its protein products. 

Almost every cell in our body has circadian oscillators [15]. The regulatory feedback loop of biological rhythm includes positive regulatory elements and negative regulatory elements. A set of core clock genes constitute a highly conserved cell-autonomous transcription-translation feedback loop (TTFL) [16], including core regulatory factors *Bmal1*, *Clock* (its homologous gene *Npas2*), *Per1*, *Per2*, *Per3*, *Cry1*, *Cry2* [17] and auxiliary factors *Rora*, *Rorb*, *Rorc*, *Dbp*, *Nfil3*, *Ciart*, *Csnk1d*, *Csnk1e*, *Dec1*, *Dec2*, *Beta-trcp*, *Fbxl3*, *Fbxl21*, *Timeless*, *Myc*, etc [18]. The basic helix-loop-helix domains (bHLH) of BMAL1 and CLOCK combine with each other to bind DNA [19,20], and both proteins contain the PAS (PER-ARNT-SIM) domain, which interacts to form an asymmetric heterodimer [21]. BMAL1/CLOCK binds to the E-box enhancer and activates the expression of *per* (*Per1*, *Per2*, *Per3*) and *Cry* (*Cry1*, *Cry2*) genes [22,23,24]. When PER and CRY in the cytoplasm continue to accumulate to a certain amount [25,26,27], both proteins are phosphorylated by casein kinase family CK1ε (encoded by *Csnk1e*) and CK1δ (encoded by *Csnk1d*) [28,29,30]. Phosphorylated PER and CRY form heterodimers and enter the nucleus, disengaging BMAL1/CLOCK from the E-box to inhibit BMAL1/CLOCK [31]. On the other hand, phosphorylated PER and CRY are ubiquitinated through E3 ligase β-TRCP (encoded by *Beta-trcp*) and FBXL3/FBXL21 (encoded by *Fbxl3/Fbxl21*) [32,33,34] and degraded through the ubiquitin-mediated pathway. In the auxiliary circuits, REV-ERBs and RORs encoded by *Nr1d1*, *Nr1d2*, and *Rora*, *Rorb*, and *Rorc* are activated when the BMAL1/CLOCK heterodimer binds to the upstream region [35,36]. REV-ERBs negatively and RORs positively regulate the expression of the *Bmal1* gene respectively [37]. In addition, DBP and NFIL3 act as activators and inhibitors, respectively, involved in regulating the transcription of D-box genes in gene promoters, such as *Per3* [38,39]. These interrelated and mutually regulated loops mediated by cis-acting elements such as E-box, RORE and D-box regions form a complex clock network. Furthermore, NPAS2 can bind to BMAL1 to form heterodimer, which also regulates the expression of clock proteins PER and CRY [40]. DEC1 (encoded by *Dec1*) and DEC2 (encoded by *Dec2*), as members of the bHLH protein family, mainly form homodimers (or heterodimers) and bind to the E-box region of the target gene promoter, thereby inhibiting gene transcription [41]. Recently, it has been discovered that CHRONO translated by *Ciart* is functionally familiar with CRY protein [42], interacts with core proteins BMAL1 and CLOCK, and plays a transcriptional inhibitory role in the negative feedback loop of transcription and translation. *Timeless* is a gene found in Drosophila that can regulate biological rhythm [43]. In mouse experiments, knockout of *Timeless* can lead to embryonic death in mice. There are currently few studies on the mammalian *Timeless* gene in the field of chronobiology. The *Myc* gene, as a new member involved in the regulation of circadian rhythms that has been studied in recent years, can down-regulate the expression of the clock-controlled gene *Bmal1* by activating the inhibition of REV-ERBs [44].

We wonder if there is a single gene in the mammalian circadian system that could determine the temperature compensation. We want to test this idea systematically. Thus, we take advantage of a powerful genetic tool, CRISPR, to knock out each individual gene that was identified as a component of the mammalian circadian clockwork. CRISPR (Clustered Regularly Interspaced Short Palindromic Repeats) is a microbial nuclease system that participates in the defense against phage and plasmid invasions [45,46]. Many types of CRISPR systems have been identified in a variety of bacteria and archaea hosts. The CRISPR/cas9 system is one of the most efficient gene-editing systems, and CRISPR/Cas9 is reconstructed in mammalian cells by optimizing the functions of endonuclease Cas9 and the essential RNA components in mammalian cells [46,47]. In addition, crRNA and tracrRNA can be fused together to produce a chimeric single guide RNA (sgRNA) [46], and Cas9 can be specifically directed to the PAM sequence through the 20-nt guide sequence in sgRNA. In view of its ease of implementation, multiplexing and other advantages, Cas9 has been used to edit eukaryotic cells and make specific mutations through NHEJ and HDR. 

In this study, we successfully knocked out each clock gene in NIH3T3 fibroblast cells [48,49]. We performed multiple Lumicycle reporter experiments on cell lines with single gene editing at 32.5 °C and 37 °C to obtain data on the circadian period, and calculated Q_10_ values for each cell line. We found that except for *Beta-trcp*-D11, the Q_10_ values of other cell lines were within a stable and normal range between 0.8 and 1.2, which indicates a complex regulatory network of temperature compensation in mammalian cells. Since temperature compensation is a basic characteristic of the biological clock and cellular homeostasis, the removal of a single clock gene can hardly affect it. Previous studies had controversial conclusions that whether the circadian cycles and the cell cycles are strongly coupled. Yeom et al. reported that the circadian system does not regulate the cell cycle rhythm in rat-1 fibroblasts [50], while Bieler et al. showed that circadian cycles and cell cycles are mutually coupled [51]. Thus, in order to understand the correlation between cell proliferation and the circadian period, we further investigated whether loss of individual clock gene would affect the cell proliferation in NIH3T3 cells. We found that the absence of clock genes such as *Clock* and *Cry* significantly prolongs cell proliferation. However, the cell growth rate is not dependent on the circadian period.

## 2. Materials and Methods

### 2.1. Cell Culture

NIH3T3-B1-B10 is an NIH3T3 cell line constructed in our laboratory with fusion expression of *Bmal1* promoter and the firefly luciferase. NIH3T3-B1-B10 cells were maintained in a complete medium (Dulbecco’s Modified Eagle Medium supplemented with 10% fetal bovine serum, and 100 U/mL penicillin-streptomycin, HyClone, Logan, UT, USA) at 37 °C (5% CO_2_). The NIH3T3 cell line was provided by Dr. Xu at Anhui University. 

### 2.2. Single Guide RNA Design

After selecting the gene to be knocked out, the exon sequence of the corresponding gene was found on the NCBI website, and the sgRNA design website of Zhang Feng’s laboratory (http://crispr.mit.edu (accessed on 10 October 2018)) was used to design the sgRNA. Then, the sgRNA sequence with the highest score and the lowest off-target rate was chosen (Appendix A). These sgRNAs were not chosen due to any specific domain of each gene. The sgRNA sequence and complementary strand were designed with *Bbs*I restriction site sticky ends and sent to company for synthesis (General Biol, Anhui, China).

### 2.3. Construction of Vectors and Cell Lines Screening

The single-stranded sgRNA sequences designed for 25 candidate genes were annealed, phosphorylated and ligated into the pX459 vector (pSpCas9{BB}-2A-Puro was a gift from Feng Zhang, Addgene plasmid No. 48139) which was digested by *Bbs*I enzyme. The constructed plasmid of each individual gene was transfected to NIH3T3-B1-B10 cells according to the instructions of Lipo6000 (#C0529, Beyotime, Shanghai, China). Twenty-four hours after transfection, cells were screened with 2 μg/mL puromycin. Next, monoclonal cell lines were screened by the limiting dilution method.

### 2.4. Identification of Knock-Out Cell Lines

After the selected monoclonal cell line growing to cell populations, we used the genome extraction kit (Tiangen, Beijing, China) to extract their genomes. We also designed the primers (Appendix A) to locate ~800 bp upstream and downstream of the target editing site to amplify the target fragment. According to the instructions of T7E1 endonuclease kit (#E001, Viewsolid, Beijing, China), the amplified target fragments were analyzed for gene editing results. Besides, the amplified target fragments were ligated into the pLB vector with subsequent sequencing.

### 2.5. Off-Target Analysis

In the sgRNA design process, the off-target site sequence and score will be given for each sgRNA (http://crispr.mit.edu (accessed on 10 October 2018)). We chose the most likely off-target site, and designed primers (Appendix A) for PCR amplification. After each single cell clone was screened for interested gene, these primers were used to amplify the possible off-target site and sent for subsequent sequencing. 

### 2.6. Bioluminescence Recording

We seeded 2 × 10^5^ cells in a 35 mm cell culture dish (#9102, Costar, Washington, DC, USA) with 2 mL medium. Replicated experiment groups and control groups were set up together. On the next day, we added 100 nM dexamethasone to synchronize the cells. Two hours later, cells were replaced with recording media without phenol red indicator, with 100 μM D-Luciferin. These dishes were covered with round cover glasses with a diameter of 40 mm. Bioluminescence data were analyzed with the LumiCycle analysis program (Actimetrics, IL, USA) to obtain circadian parameters such as period and amplitude. 

### 2.7. Immuno-Blotting Assays

Mouse antibodies against Myc-tag (#M047-3, MBL, Japan), rabbit antibody against GFP-tag (D110008, Sangon, Shanghai, China) and Rabbit antibody against HistonH3A (#D151007, Sangon, Shanghai, China) were subjected to Western blot according to the manufacturer’s protocol. For WB, HEK-293T cells transfected with the desired plasmids by using Lipo8000 Transfection Reagent (Beyotime, Beijing, China). Cells were harvested in RIPA buffer (2 M Tris pH 8.0, 5 M NaCl, 0.5 mM EDTA, 1% NP-40, 10% glycerinum). After quantification by using BCA Protein Assay Kit (P0012, Beyotime, Beijing, China), lysates were added to the 5 X SDS loading dye, boiled for 5 min, and run on a 12.5% SDS-PAGE gel followed by Western blot analysis. Immunoreactive bands were detected by Typhoon laser scanners (FLA-9500, GE, Boston, MA, USA).

### 2.8. Cell Proliferation Assay

The cell suspension was diluted to keep a cell concentration of 2000 cells/mL and was seeded to a 96-well plate with 200 μL per well. A zero-setting well (medium only without cells) and a control well were set up at the beginning. For each cell line, 5 groups of duplicate control groups were set, and 6 plates of the same cells were plated simultaneously. After the cells adhered to the bottom, we added 10 μL of 5 mg/mL MTT reagent to all the experimental and control wells, including the zero-setting well, and then wrapped the 96-well plate with tin foil and put it back into the cell culture incubator for four hours. Next, we aspirated the medium carefully to avoid touching the crystals at the bottom of the well. 150 μL of DMSO was added to each well and the plate was shaken at high speed for 30 min in a shaker to ensure that the blue crystals were fully dissolved. The absorbance of each well was measured at a wavelength of 490 nm by a multifunctional microplate reader.

24 h later, the absorbance value of the second plate of cells was measured by the same method. The third plate of cells was measured 36 h later. Notably, the medium in the remaining three untreated culture dishes was replaced with fresh media. One plate of cells was selected for measurement every other day until the six plates of cells were completely measured. According to the collected six sets of data, the cell doubling curve was drawn with time, days as the abscissa and OD value as the ordinate. The SPSS software was used to fit the two-division data to calculate the cell proliferation rate.

## 3. Results

### 3.1. Cellular Circadian Gene Bmal1 Deletion by the CRISPR System Resulted in Loss of Circadian Rhythms

To understand the influence of each individual clock gene on the cellular physiology, we used CRISPR/Cas9 system to edit 23 clock genes plus *Myc* and *Timless* in NIH3T3-B1-B10 cells. This cell line can stably express luciferase, which is driven by a *Bmal1* promoter, so as to observe the circadian rhythm. Taking *Bmal1* as an example, the procedure of gene editing using CRISPR tools is presented (Figure 1). The single guide RNA sequence was designed and constructed to the pX459 vector as described in the Material & Methods (Figure 1A). A stable *Bmal1* deletion cell line, *Bmal1*-D8, was established after normal procedures. Sequencing analysis identified that both alleles of *Bmal1* were mutated by the CRISPR technique (Figure 1B). One allele had a deletion of 125 bp leading to a premature termination, while the other allele had a deletion of 2 bp (Figure 1B and Appendix A). Next, we asked if the deletion of *Bmal1* will eliminate the circadian rhythms. We did the LumiCycle reporter assay to monitor the circadian rhythms, and the results showed that the knockout of *Bmal1* resulted in a loss of the circadian rhythm (Figure 1C,D).

In order to prove that the phenotype is indeed caused by the deletion of *Bmal1*, we carried out complementation experiments. First, a lentiviral complement vector was constructed, which contains a *Bmal1* encoding box under the CMV promoter. We used 5xMyc as a tag fused to the N-terminus of Bmal1, in order to detect the complemented protein (Figure 1E). By western blotting studies, we confirmed that this lentiviral construct can successfully express exogenous BMAL1 in various cell lines including *Bmal*-D8 cells (Figure 1F). However, expression of exogenous *Bmal1* under the CMV promoter failed to rescue the circadian rhythm in *Bmal*-D8 cells (Figure 1G). 

### 3.2. The Expression of Exogenous Bmal1 under a Bmal1 Promoter Can Rescue the Rhythm in Bmal-D8 Cells

CMV is a strong constitutive promoter, which may explain why the circadian rhythm of *Bmal1*-D8 has not been restored. Next, we constructed another lentiviral vector (Figure 2A). This new lentiviral construct contains the *Bmal1* promoter which drives the expressions of EGFP and Flag/HA tagged BMAL1 proteins simultaneously. In this construct, a ribosome binding site IRES was introduced between EGFP and the interested gene to ensure EGFP and the interested protein, BMAL1, can express at the same time (Figure 2B,C). It was found that the circadian rhythm was restored after expressing the exogenous BMAL1 under *Bmal1* promoter (Figure 2D), but the robustness of the rhythm did not return to the level of wild type, possibly because these are mixed cells. The *Bmal1*-D8 is successfully rescued, as the FLAG tag on the expressed BMAL1 can be detected using western blotting (Figure 2F). Moreover, the circadian period of *Bmal1*-D8 with complemented FH-BMAL1 recovered to the same extent as that of the wild-type cells (Figure 2E).

### 3.3. Systematic Knocking out Clock Genes Using CRISPR/Cas9

Taking *Bmal1* as an example, the loss of circadian rhythm in *Bmal1*-D8 cells showed that CRISPR/Cas9 is an effective tool to knock out clock genes. We would like to further knock out circadian genes that will show period changes. Based on previous knowledge, loss of *Cry1* or *Cry2* genes will shorten or lengthen the circadian periods, respectively [52]. Thus, we choose *Cry1* and *Cry2* genes to further verify the effects of knocking out genes by the CRISPR/Cas9 tools. As illustrated in Figure 3A,B, sgRNAs targeting the exon 1 of either *Cry1* or *Cry2* genes were chosen and constructed into pX459 vectors to knock out corresponding genes. After screening for successful knock-out cell lines, we obtained *Cry1*-C9 and *Cry2*-F12 respectively. The period estimations using Lumicycle reporter assays showed that *Cry1*-C9 cells have a significantly shortened period by 0.48 h while *Cry2*-F12 cells have a significantly lengthened period by 1.1 h (Figure 3C, Table 1). These circadian period phenotypes are comparable with those of gene knock-out animals [52]. Therefore, CRISPR/Cas9 tools are reliable to knock out each individual gene. 

Next, we designed sgRNAs targeting known clock genes and transfected each construct to NIH3T3-B1-B10 cells. Except for *Nr1d1*, *Myc*, and *Timeless* genes, we obtained knock-out cells of other genes mentioned in the Introduction (Table 1 and Appendix A). The comparison of circadian period changes between knockout mice and knockout cell lines demonstrates that most of the period phenotype changes are consistent for the same gene, indicating that the circadian system at cellular level is the same as the animal level. There was no available information about the circadian phenotypes of *Nr1d2* or *Nfil3* knockout animals, and we obtained the knockout cell lines of these two genes. There are a few conflicting results between knockout animals and knockout cell lines, for example, *Csnk1d*^−/+^ showed a shortened period, while *Csnk1d*-D2 showed a lengthened period (Figure 3C, Table 1). However, the animal study of *Csnk1d* was a heterozygous animal. In general, knocking out an individual circadian gene did not exhibit a dramatic effect on circadian period.

### 3.4. Temperature Compensation Evaluated in Cells Which Have Systematically Knocked out Clock Genes

According to preliminary explorations of the circadian oscillation of NIH3T3-B1-B10 at different temperatures (data not shown), we selected 32.5 °C and 37 °C in order to record robust circadian oscillation of cells under physiological range. The circadian period at different temperatures was measured by the Lumicycle luminometer and the effect of gene editing on temperature compensation was analyzed by calculating the numerical value of Q_10_ based on the following equation. τ1 and τ2 represent the circadian period at temperature T1 and T2, respectively (T1 = 32.5 °C, T2 = 37.0 °C).
(1)$$Q_{10}={\left(\frac{\tau 1}{\tau 2}\right)}^{\frac{10}{T2-T1}}$$

By calculating the Q_10_, we found that knocking out each single clock gene had little effect on temperature compensation. For all the cell lines we obtained, the Q_10_ value ranges from 0.8 to 1.2, except of *Beta-trcp*-D11 cell line (Figure 4). We observed that there are two categories in which the circadian period changes with the increase of temperature. One category includes *Cry1*-C9, *Cry1*-D4, *Per1*-B7, *Per1*-C1, *Per3*-D4, and *Fbxl3*-A9, whose periods become shorter with the increase of temperature, while the circadian periods of the other category become longer (Figure 4). We wondered if different types of the clock genes show different trends in terms of temperature compensation. Thus, we plot the Q_10_ values based on the types of these clock genes: core positive regulator (Figure 4A), core negative regulators (Figure 4B,C), *Bmal1* gene regulators (Figure 4D), E3 ligases (Figure 4E), auxiliary regulators (Figure 4F), and casein kinase genes (Figure 4G). However, the Q_10_ is relatively consistent in all types. Thus, we conclude that the existence and maintenance of temperature compensation is regulated by complex gene networks rather than single clock genes. In addition, the Q_10_ value of *Beta-trcp*-D11 is less than 0.8. We preliminarily concluded that *Beta-trcp* may be more related to the maintenance of temperature compensation since it is involved in the stability of PER proteins. Though a dramatic change in Q_10_ was observed in *Tau* (CK1ε-R178C) mutant animals [10], *Csnk1e*-B5 cells showed little effect on Q_10_. Thus, not like KaiC in cyanobacteria, key kinases in the mammalian clock system do not determine the characteristics of temperature compensation. 

### 3.5. Proliferation Rates Evaluated in Cells Which Have Systematically Knocked out Clock Genes

Since we have observed different circadian period changes of these knocked-out cell lines, we planned to investigate whether there is a correlation between circadian period change and the cell proliferation rate change. We used MTT colorimetric method to determine the cell growth curve. According to the six groups of data collected, the cell doubling curve was plotted based on the recorded OD values (Figure 5A). We inoculated the same number of cells on the first day, and all the cell lines were maintained in logarithmic growth phases within six days. Again, we plot the growth curve of these cell lines based on different roles these genes play in the circadian clock networks. Next, we used SPSS software to calculate the doubling time of cell lines with two-division mode Y = a × 2^bt^ (Figure 5B). We found that the doubling time of most gene-edited cell lines was not different from that of the wild type. Nevertheless, the doubling time of *Clock*-F12, *Cry1*-D4, *Per1*-B7, *Per3*-D4, *Nr1d2*-F4, and *Csnk1e*-B5 were significantly prolonged, while that of *Per2*-A3 was significantly shortened. However, the mechanisms underlying the changed proliferation rates need to be further studied. Taken together, by systematically comparing the circadian period and proliferation rates of these cell lines, it showed that no strong correlation can be found between the circadian period and the cell proliferation (R^2^ = 0.0998, *p* value = 0.1086, Figure 5C). 

## 4. Discussion

In this study, we reported a systematic investigation on the temperature compensation and cell growth rate over currently known mammalian circadian genes in a fibroblast cell line. We applied a powerful genetic manipulation tool, CRISPR/Cas9, to generate a series of mutant cell lines that have individual circadian gene being knocked out (Table 1 and Appendix A). Most of these cell lines showed robust circadian oscillations, except the *Bmal1*-D8 and *Clock*-F12 cell lines. *Bmal1* gene was taken as an example to verify the gene deletion by CRISPR/Cas9. Deletion of *Bmal1* resulted in loss of the circadian rhythm, while complementation of exogenous *Bmal1* under the *Bmal1* promoter can rescue the rhythm. Most cell mutants recapitulate the circadian phenotypes of the period changes, which were collected using knockout animal studies. For example, *Cry1*-C9 cells showed a half-hour shortened period, and *Cry2*-F12 cells showed one hour longer period (Table 1, and Appendix A). Therefore, our mutant cells represent the first full collection of knockouts of clock genes. According to our sequencing results of the second-high possibility of potential off-target gene locus (Appendix A), there were no off-target mutations in this study. However, we cannot exclude the possible off-target mutations in the less potential loci.

Our data demonstrated that the period length of circadian rhythms is temperature compensated in each mutant cell line, except the arrhythmic ones, *Bmal1*-D8 and *Clock*-F12. Although previous results indicate that CK1δ/ε might be the important factor that determines the temperature compensation in the mammalian system [11,12], our data show that CK1δ and CK1ε depletion cells are still temperature compensated, with Q_10_ value close to that of the wild type cells (Figure 4G). This indicates that temperature compensation is not determined by a single factor. A single factor might be the key in other model organisms, such as *frq* in neurospora [9] and *KaiC* in cyanobacteria [12,13]. However, we observed a few mutant cells which showed a different trend in terms of period changes at different temperatures, 32.5 °C and 37 °C. For instance, *Cry1*-C9 showed a longer period at lower temperature, while WT and most other mutant cells showed a shorter period at lower temperature (Figure 4). We sort all known circadian genes into several categories, positive transcription factors, negative regulators, *Bmal1* gene regulators, E3 ligases leading to degradations, auxiliary regulators and kinases. Then we compared the period change in these different categories. The period-got-increased trend appeared differently in these categories, indicating that this phenomenon might be gene-specific, but not dependent on the properties of a specific category. Next, we considered the effect of period change on the trend of temperature compensation. In the mutant cells that have shorter period, *Cry1*-C9, *Per1*-C1and *Per3*-D4 (Figure 3C), the trends are opposite to those of other cell lines (Figure 4). This is a very interesting point, which means that cells with slower clock at lower temperature (32.5 °C) can run faster at higher temperature (37 °C). In most cases, for example, wild-type cells, cells’ clock runs faster at lower temperature and runs slower at higher temperature, resulting in Q_10_ value less than one (Figure 4). These data indicate that the temperature compensation is a fundamental property of the circadian clock which is maintained in a homeostatic way under physiological temperature ranges. 

Thus, we conclude that the temperature compensation is not determined by properties of each gene product, nor the properties of any limb in the circadian system. We speculate that the temperature compensation might be achieved by a network of multiple genes, which need further studies. Not like the circadian protein KaiC, which shows temperature compensated kinase and ATPase activities in pure biochemical reactions, none of the known mammalian clock gene products possess temperature compensation. Casein kinase 1ε has been implied to play important role in temperature compensation [10], but knock-out of CK1ε showed similar Q_10_ to that of WT cells (Figure 4G). Other studies have tried to double knock out CK1δ and CK1ε but failed to get target cells, which indicate at least one allele of CK1δ/ε is essential for cell growth [67].

We also investigated the relation between circadian period and cell growth rate. Compared to the changes of the circadian period of these mutant cells, the changes of the growth rate are larger in some of the cells (Figure 5B). Our results demonstrate that the cell growth rate is not dependent on the circadian period. Previous studies indicate that the circadian cycle gates the cell cycle, but we showed that the growth rate of most mutant cells is independent of the lengthened period nor shortened period. It may be possible that the circadian cycle gates the cell cycle, but the non-circadian functions of these genes may play roles at different phases of cell cycles. In order to fully understand the relation between circadian rhythms and cell cycle, further study is necessary.

## 5. Conclusions

In conclusion, we knocked out the known circadian genes in mouse fibroblasts and found that their biological rhythms are not destroyed except Bmal1-D8 and Clock-F12 cell lines. We concluded that the temperature compensation is a complex gene regulation system instead of being regulated by any single gene. The temperatures we used in this study are limited to 32.5 °C and 37 °C. Although the bioluminescence rhythm at 25 °C was not consistent in our hands, more temperatures are required to test in the near future. Furthermore, we found that the cell growth rate is not dependent on the circadian period by analyzing and comparing the proliferation rates and circadian periods of knockout cell lines.

## Figures and Tables

**Figure 1 biology-10-01204-f001:**
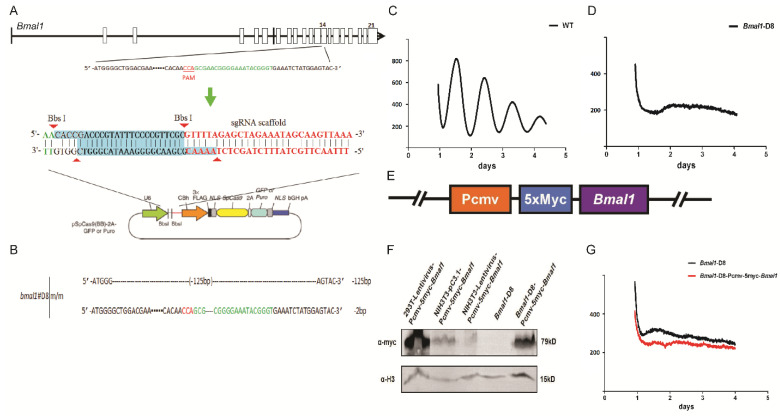
NIH3T3 cells with *Bmal1*^−/−^ genotypes are arrhythmic. (**A**) The schematic diagram of sgRNA design information and knockout vector construction of the *Bmal1* gene is shown. On the top, the schematic of *Bmal1* was drawn. In the enlarged region at the 14th exon, the bases marked in green are the sequence of the picked sgRNA. Then, the sgRNA sequence was cloned into the px459 vector. (**B**) A monoclonal cell line, *Bmal1*-D8, showed that both alleles of the *Bmal1* gene were edited. According to the analysis on the predicted mRNA, neither edited alleles of *Bmal1* is capable of producing functional Bmal1 proteins. (**C**) The luminescence recording of the WT NIH3T3 cells that bear with a luciferase reporter under the *Bmal1* promoter. (**D**) The bioluminescence rhythm of the *Bmal1*-D8 cells, which is arrhythmic. (**E**) Illustration of the main components of the constructed lentiviral vector to complement the *Bmal1* gene. The Bmal1 coding sequence tagged with five constitutive myc tags is under the CMV promoter. (**F**) Myc epitope tags as a label to detect whether the cells are successfully transfected with exogenous *Bmal1*. In several cell lines, 293T, NIH3T3, and *Bmal1*-D8 cells, lentiviral constructs can successfully express Bmal1 proteins. Histone3 is shown as a loading control. Histone3 is shown as a control. (**G**) However, both *Bmal1*-D8 and *Bmal1*-D8 cells supplemented with lentivirus were arrhythmic. Representative luminescent trace or image was shown.

**Figure 2 biology-10-01204-f002:**
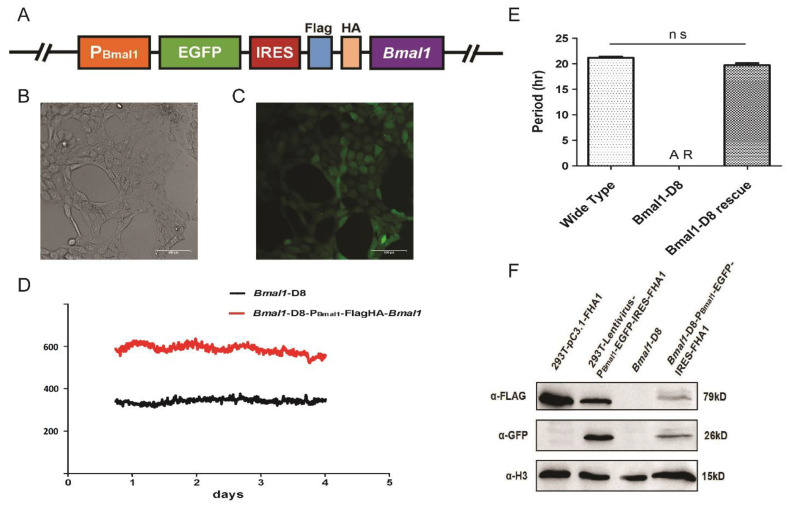
The lentivirus of exogenous *Bmal1* driven by the *Bmal1* promoter rescues the circadian phenotype of *Bmal1*-D8. (**A**) Illustration of the main components of the new constructed lentiviral complement vector, in which the Flag-HA tagged *Bmal1* is driven by the native *Bmal1* promoter. (**B**) *Bmal1*-D8 cells were transfected with the new lentiviral vector and the bright filed image is shown. (**C**) The fluorescent image of *Bmal1*-D8 cells transfected with the new lentiviral vector indicates that GFP is successfully expressed. (**D**) The bioluminescence rhythm results of *Bmal1*-D8 and lentivirus infected *Bmal1*-D8-P_Bmal1_-FlagHA-*Bmal1* cells. (**E**) The calculated circadian period of wide type, *Bmal1*-D8, *Bmal1*-D8 rescue cells. Bmal1-D8 rescue cells showed a period closed to that of the wild-type cells. (**F**) Western blot images show that the GFP and Flag-HA tagged *Bmal1* can simultaneously express in both HEK293T and *Bmal1*-D8 cells. Histone3 are shown here as loading controls. ns, none significant. Representative luminescent trace or image was shown.

**Figure 3 biology-10-01204-f003:**
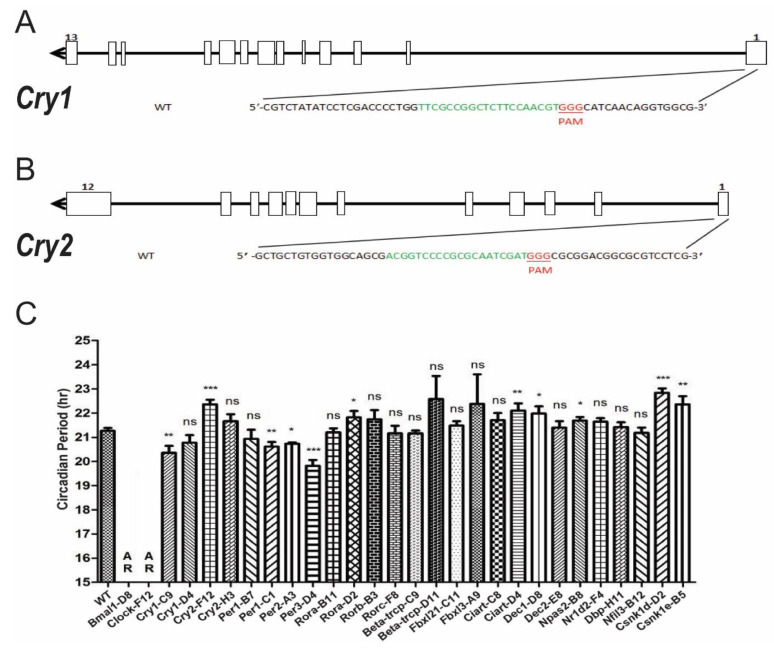
Circadian period phenotypes of cell lines with individual clock gene being knocked out. (**A**) The schematic diagram of sgRNA designed to knock out the *Cry1* gene in NIH3T3 cells. (**B**) The schematic diagram of sgRNA designed to knock out the *Cry2* gene in NIH3T3 cells. (**C**) The statistical results of the circadian period of the monoclonal cell lines for the clock genes in this study. Similar to the animal studies, *Cry1* deletion cell lines showed shortened period, while *Cry2* deletion cell lines showed lengthened period. The period changes were summarized in Table 1. Results are expressed as mean ± SD, n ≥ 3 for each group. ns, no significant, * *p* < 0.05, ** *p* < 0.01, *** *p* < 0.001.

**Figure 4 biology-10-01204-f004:**
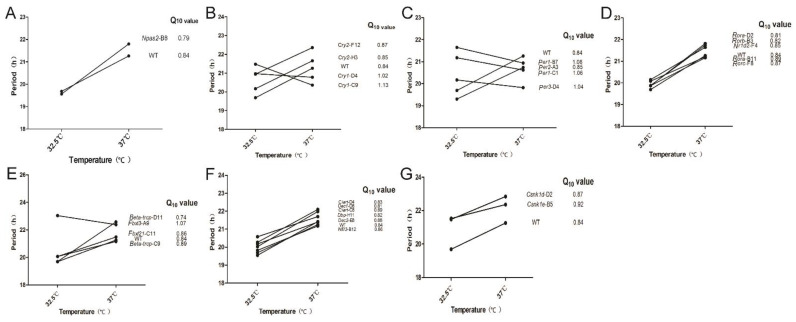
The circadian period changes of monoclonal cell lines at different temperatures. Clock genes in this study were classified into different types. (**A**) The circadian period changes of wild-type and *Npas2*-B8 cells at different temperatures. Since *Bmal1* deletion and *Clock* deletion cells were arrhythmic, *Npas2* is shown here to represent the core positive regulator group. (**B**) The circadian period changes of *Cry1*-D4, *Cry1*-C9, *Cry2*-F12, *Cry2*-H3 cells at different temperatures. (**C**) The circadian period changes of *Per1*-B7, *Per1*-C1, *Per2*-A3, *Per3*-D4 cells. *Cry* and *Per* genes are core negative regulators. (**D**) The circadian period changes of *Rora*-B11, *Rora*-D2, *Rorb*-B3, *Rorc*-F8, *Nr1d2*-F4 cells, which represent regulators for the *Bmal1* gene. (**E**) The circadian period changes of *Beta-trcp*-C9, *Beta-trcp*-D11, *Fbxl21*-C11, *Fbxl3*-A9 cells at different temperatures. These genes represent E3 ligases. (**F**) The circadian period changes of *Ciart*-C8, *Ciart*-D4, *Dec1*-D8, *Dec2*-E8, *Dbp*-H11, *Nfil3*-B12 cells, which represent auxiliary regulators. (**G**) The circadian period changes of the casein kinase genes, *Csnk1d*-D2, *Csnk1e*-B5 cells at different temperatures. On the right is their Q_10_ value. Periods of each cell line were calculated based on multiple experiments.

**Figure 5 biology-10-01204-f005:**
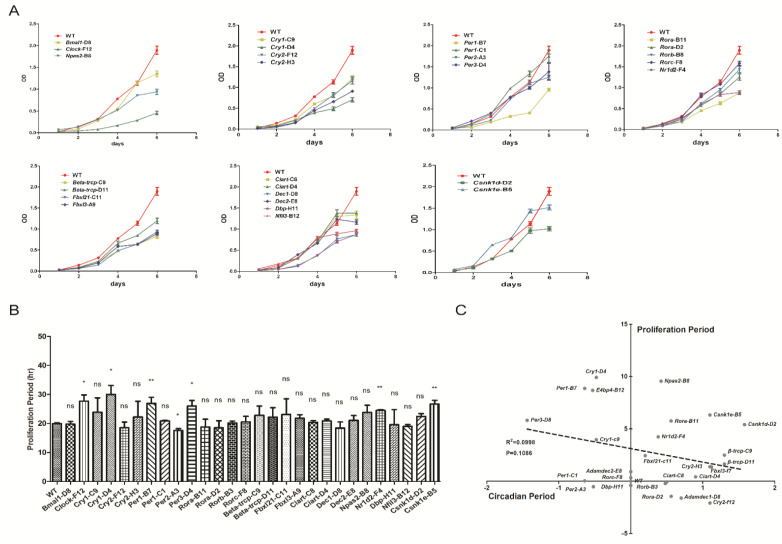
The proliferation periods of clock gene mutant cells are not correlated with their circadian periods. (**A**) Cell proliferation curves of wild-type and clock gene deletion monoclonal cell lines for six consecutive days. (**B**) The proliferation period of each cell line was calculated and plotted. (**C**) The relationship between the circadian period changes and the proliferation period of the monoclonal cell lines. Linear relationship was applied to fit the data in panel C. Results are expressed as mean ± SD. n ≥ 3 in each group. ns, no significant, * *p* < 0.05, ** *p* < 0.01.

**Table 1 biology-10-01204-t001:** Comparison of rhythm cycle changes between knockout mice and cell lines.

Genes	Alleles	Mouse Mutant Phenotypes	Cellular Mutation Phenotypes
*Bmal1*	*Bmal1* ^−/−^	Rhythm disorder [39]	Rhythm disorder
*Clock*	*Clock* ^∆19/^ ^∆19^	Rhythm disorder [39]	Rhythm disorder
*Npas2*	*Npas2* ^−/−^	Shorten 0.2 h [53]	Lengthen 0.42 h
*Cry1*	*Cry1* ^−/−^	Shorten 0.5 h [52]	Shorten 0.48 h
*Cry2*	*Cry2* ^−/−^	Lengthen 1 h [52]	Lengthen 1.1 h
*Per1*	*Per1* ^−/−^	Shorten 0.5 h [54]	Shorten 0.64 h
*Per2*	*Per2^brdml^*	Shorten 1.5 h [55]	Shorten 0.52 h
*Per3*	*Per3* ^−/−^	Shorten 0–0.5 h [56]	Shorten 1.44 h
*Rora*	Staggerer	Shorten 0.5 h [35]	Lengthen 0.56 h
*Rorb*	*Rorb* ^−/−^	Lengthen 0.5 h [57]	Lengthen 0.48 h
*Rorc*	*Rorc* ^−/−^	WT [58]	WT
*Nr1d1*	*Nr1d1* ^−/−^	Shorten 0.5 h [59]	No cell lines
*Nr1d2*	*Nr1d2* ^−/−^	N/A	Lengthen 0.38 h
*Dbp*	*Dbp* ^−/−^	Shorten 0.5 h [60]	WT
*Nfil3*	*Nfil3*^−/−^Nfile3	N/A	WT
*Ciart*	*Ciart* ^−/−^	Lengthen [61]	Lengthen 0.5–0.9 h
*Dec1*	*Dec1* ^−/−^	Lengthen 0–0.15 h [62]	Lengthen 0.7 h
*Dec2*	*Dec2* ^−/−^	WT [62]	WT
*Beta-trcp*	*Beta-trcp* ^−/−^	WT [63]	WT/Lengthen 1.3 h
*Fbxl21*	*Fbxl21* ^−/−^	WT [33]	Lengthen 0.2 h
*Fbxl3*	*Fbxl3* ^−/−^	Lengthen 2–3 h [32]	Lengthen 1.12 h
*Myc*	*Myc* ^−/−^	Lengthen [44]	No cell lines
*Timeless*	*Timeless* ^−/−^	Embryo death [64]	No cell lines
*Csnk1d*	*Csnk1d* ^−/+^	Shorten 0.5 h [65]	Lengthen 1.58 h
*Csnk1e*	*Csnk1e* ^−/−^	Lengthen 0.3 h [66]	Lengthen 1.1 h

N/A: Not available.

## Data Availability

Not applicable.

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
