# Peer review of "Systematic Studies of the Circadian Clock Genes Impact on Temperature Compensation and Cell Proliferation Using CRISPR Tools"

_biology, 2021, doi:10.3390/biology10111204_

Round 1

Reviewer 1 Report

The temperature-compensated circadian clock is well established in the literature. Accordingly, the authors attempted to understand the mechanism of temperature compensation by creating 22 individual circadian clock cell lines, each with a different circadian clock gene knocked out using a CRISPR KO strategy. The authors report that they did not observe any adverse effect on temperature compensation by any cell line, therefore they concluded that temperature compensation is not gated through the circadian clock, but rather by cellular homeostasis instead of a single gene. Additionally, the authors found that cell proliferation is not dependent on the circadian period despite the reports in the literature that clearly show the effects of circadian proteins on cellular proliferation with mechanistic links to bridge the two together. 

There are major and minor concerns, centering on the poor quality of the data and lack of inclusion of raw rhythms. Overall, the manuscript is poorly written and requires significant improvement. 

  1. The authors must show the rhythm results of each and all the KOs that serve as evidence that the KOs are functionally validated. 
  2. It is not clear if the authors used a random search for sgRNAs or designed sgRNAs in specific domain regions of each sequence.
  3. Previous temperature compensation studies have observed clock function at 24oC, 27oC, 30oC, and 37oC. The authors did not compare their results to previous findings. It is possible that more drastic effects may be seen at different temperateness and that they missed those effects by limiting this study to 32.5oC and 37oC. Comparing 24oC, 30oC, and 37oC may have been more informative and the results from the study could be easily compared to those in the literature.
  4. Links between the circadian clock and the cell cycle have been well established. Furthermore, there is literature on the influence of the clock on cell proliferation that contradicts the authors' findings.
  5. Lumicycle data: must define and show values for luminescence. 
  6. The authors state that Nr1D2 and Nfil3 have not been knocked out in an animal. Incorrect.
  7. Fig 1. Western blot is of low quality and not trustworthy.
  8. Fig 2. I do not see rescue by Bmal1.
  9. The rhythm data for all Figures especially 3 and 4 must be presented.
  10. The manuscript was poorly written and requires significant rewriting.

Author Response

Please see the attachment for our reply to Reviewer 1. 

Reviewer 2 Report

The authors demonstrated the effects of knock-out of circadian clock genes on temperature compensation and cell proliferation. The idea of the study is interesting. However, the manuscript needs certain improvements. The reviewer has a few comments which the authors may address: 

  1. The authors are suggested to explain their rationale in detail. And discuss if there is any connection between temperature compensation and cell proliferation.
  2. The authors are suggested to introduce the temperature compensation and the definition of Q10 which would help the readers understand the importance of the study.
  3. The authors are suggested to introduce some information about Cry1 or Cry2 genes to help readers understand the paper.
  4. Western blot should be included in the part of “Materials and methods”.
  5. Figure 1F needs to be reproduced due to the quality.
  6. The authors should discuss the cell-line Beta-trap-D11 in detail since only this cell-line has a Q10 value less than 0.8.
  7. The authors are suggested to detect the expression level of each knock-out gene and present the WB results in the supplementary material.
  8. The author should give a brief description for conclusion. The author should give the critical summaries. Conclusion should provide data of key findings, novelty of work and applicability.

Author Response

Please see the attachment for our reply to Reviewer 2.

Round 2

Reviewer 1 Report

My concerns were not sufficiently addressed, but the reagents developed in the study may be useful to others in the field. Therefore, I recommend acceptance for publication. 

Reviewer 2 Report

In the revised article, the authors modified the manuscript and figures referred to the comments, answered the questions comprehensively.